# Real-time imaging of sulfhydryl single-stranded DNA aggregation

Fanwei Zeng[1,5], Youhong Jiang[1,5], Nana He[1], Tiantian Guo[2], Tiqing Zhao[1], Mi Qu[1], Yue Sun[1], Shuting Chen[1], Dan Wang[3], Yong Luo[3], Guangwen Chu[3], Jianfeng Chen[3], Shi-Gang Sun [1] & Hong-Gang Liao [1,4✉]

The structure and functionality of biomacromolecules are often regulated by chemical bonds, however, the regulation process and underlying mechanisms have not been well understood. Here, by using in situ liquid-phase transmission electron microscopy (LP-TEM), we explored the function of disulfide bonds during the self-assembly and structural evolution of sulfhydryl single-stranded DNA (SH-ssDNA). Sulfhydryl groups could induce self-assembly of SH-ssDNA into circular DNA containing disulfide bonds (SS-cirDNA). In addition, the disulfide bond interaction triggered the aggregation of two SS-cirDNA macromolecules along with significant structural changes. This visualization strategy provided structure information at nanometer resolution in real time and space, which could benefit future biomacromolecules research.

---

[1] State Key Laboratory of Physical Chemistry of Solid Surfaces, Collaborative Innovation Center of Chemistry for Energy Materials, College of Chemistry and Chemical Engineering, Xiamen University, Xiamen, China. [2] Fujian Provincial Key Laboratory of Neurodegenerative Disease and Aging Research, Institute of Neuroscience, School of Medicine, Xiamen University, Xiamen, China. [3] State Key Laboratory of Organic-Inorganic Composites and Research Center of the Ministry of Education for High Gravity Engineering and Technology, Beijing University of Chemical Technology, Beijing, China. [4] Innovation Laboratory for Sciences and Technologies of Energy Materials of Fujian Province (IKKEM), Xiamen, China. [5]These authors contributed equally: Fanwei Zeng, Youhong Jiang. ✉email: hgliao@xmu.edu.cn

Disulfide bonds play an important role in numerous natural biomacromolecules and artificial biomaterials[1]. These bonds are involved in folding and stabilizing the secondary structure of proteins and controlling the self-assembly of tertiary structures between different subunits in proteins. Any disruption of these structures is strongly associated with loss of protein function and activity[2–4]. As a functional group, the disulfide bond has been widely used to develop drug delivery and release systems because of its characteristics of self-assembly and intracellular glutathione-responsive destabilization[5,6].

However, the process and mechanism of disulfide bond formation and rearrangement in biomacromolecules need to be explored. Many technologies, including atomic force microscopy, computer simulation, and chemical color-reaction method, have been used to study disulfide bond formation and rearrangement[7–10]. For the absence of real-time imaging of the dynamic process at nanoscale, detailed information on the structural changes and rearrangement induced by disulfide bonds is lacking.

In situ liquid-phase transmission electron microscopy (LP-TEM) is now being used to study material transformation dynamics in liquids, and several studies shown the formation and self-assemble of nanoparticles in liquids[11–15]. The liquid environment can protect biomacromolecules, such as proteins and DNA, from electron beam damage[7,16]. The observation of DNA in liquid has also been demonstrated by using LP-TEM[16,17].

In this study, we used sulfhydryl single-stranded DNA (SH-ssDNA) as a model to investigate the self-assemble and rearrangement of biomacromolecules controlled by disulfide bonds in LP-TEM. ssDNA modified by sulfhydryl groups could self-assemble into irregular and unstable nano-agglomerates. By leveraging LP-TEM, we investigated SH-ssDNA dynamics triggered by disulfide bonds to understand these chemical bonds and their functions in biomolecules.

## Results and discussion

### Self-assembly behavior of ssDNA and SH-ssDNA.
In order to capture the dynamics of macromolecules with more accurate and effective, we chose ssDNA as a model to carry out the following studies, since the deoxynucleotide possess higher molecule size and contrast in liquids than amino acids. Here, we tracked the self-assembly behavior of ssDNA and SH-ssDNA at 5 μM concentrations. The electron beam dose rate distribution reported in previous articles ranges from 0.21 to 100 $e^-Å^{-2}s^{-1}$[16,18–20]. Here we use 60 $e^-Å^{-2}s^{-1}$ as an intermediate dose. And we adopt PBS system as solvent for DNA, which has high buffering capacity. The $H^+$ and $OH^-$ produced by electron beam irradiation will be neutralized and consumed, thus reducing radiation damage (Fig. 1). ssDNA could self-curl into nanoclusters in the solutions (Fig. 2a and Supplementary Video 1), and the average diameter was ~1.4 nm (Fig. 2b). Compared with metal nanoparticles[13,14], the boundary of the nanocluster here was unclear and the contrast was low. However, when the image was enlarged, ssDNA had a thin string-like structure (Fig. 2a). Because of the resolution limit in liquid, the base sequence and specific shape of ssDNA presented obscure visual effectiveness, and the specific shape may have improved using single-particle modeling and image simulation. In addition, compared with cryo-electron microscopy, the surface morphology observation of single particles showed no obvious differences[21].

To study the influence of the sulfhydryl group on ssDNA self-assembly, we prepared SH-ssDNA with two sulfhydryl bonds (-SH) modified at the terminal of ssDNA. Compared with ssDNA, SH-ssDNA showed a clear shape (Fig. 2a). In the 5 μM SH-ssDNA solution, SH-ssDNA showed the regular circular shape with middle gaps and was thick string-like structure after zooming in Fig. 2a. This observation also showed that SH-ssDNA could self-assemble into circular shaped DNA containing disulfide bonds (SS-cirDNA)[17,22], and the average diameter of the macromolecule was 2.4 nm (Fig. 2b and Supplementary Video 2). Interestingly, the macromolecules rotated continuously, and the mean square displacement curve follows an approximate linear behavior, as published report[23], and the slope amounted to 0.1025 nm²/s, showed characteristics of Brownian motion (Supplementary Fig. 1). The average distance between macromolecules was 5.4 nm, and the size of macromolecules ranged from 1.2 to 3.4 nm (Fig. 2b). These results could be ascribed to the two "-SH" of the sulfhydryl-bond terminated ssDNA forming a disulfide bond (-S-S-). Thus, SH-ssDNA self-curled into circular-shaped SS-cirDNA, with single SH-ssDNA connected end-to-end or multiple SH-ssDNA automatically combined, which is consistent with published results[24,25].

Considering the electron beam effect and surface adsorption in the environment of LP-TEM[9,13], we carried out the self-polymerization experiment of SH-ssDNA in bulk solution to exclude the electron beam effect and detected the result by Nucleic acid electrophoresis experiments. The result showed that compared with ssDNA, the molecular weights of SH-ssDNA sample is significantly larger than that of ssDNA sample, which means that polymerization occurs between SH-ssDNA. What's more, the molecular weights distribution of the SH-ssDNA sample ranging from 100 bp to 5000 bp indicates many different polymers are formed (Fig. 2c). These results suggested that the polymerization of SH-ssDNA induced by sulfhydryl groups is spontaneous rather than by electron beams effect or surface adsorption. However, better resolution of LP-TEM will help decide on the degree of disulfide bond participation is involved.

To understand the influence of concentration on SH-ssDNA assembly behavior, we prepared 0.5 and 5 μM SH-ssDNA solutions. Interestingly, the SH-ssDNA molecules presented different states at different concentrations (Fig. 3a, Supplementary Videos 3 and 4). At low concentration, the SH-ssDNA molecules were mainly small multimers, but they have a tendency to form larger multimers in the 5 μM SH-ssDNA solution. Along with the larger multimers, macromolecule size showed an increasing trend, whereas macromolecule number showed a decreasing trend (Fig. 3b, c). In contrast, at low concentrations, SH-ssDNA remained in the oligomer state (Fig. 3b, c). We further observed that the larger particles showed circular structure with middle gaps, while the smaller particles showed cluster structure with no voids in the center. According to the statistics of circular particles and cluster particles respectively, it was found that compared with 0.5 μM SH-ssDNA, there were more circular particles in the group of 5 μM SH-ssDNA (Fig. 3d), which might be due to the increase of concentration, and then increase the chance of particle collision and aggregation. These results showed assembly behavior of SH-ssDNA is concentration dependent.

### The effects of electron-sample interactions and surface adsorption.
The possibility of damage caused by the electron beam have to be accounted during the in situ liquid phase TEM test. Water molecule would be radiolysis by the electrons and generated a few species, including hydrated (solvated) electrons $e^-$(aq), hydrogen radical H•, hydroxyl radical OH•, and $H_2$. In our experiment, the electron does is 60 $e^{-1} Å^{-2} s^{-1}$. According to the reference[26], the concentration of the above species is around $10^{-7}$ to $10^{-5}$ mol/L, which is a fairly low concentration and the effect on the reaction between SH-DNA is negligible. The parallel electron beam interacts with DNA and this might disturb the dynamics of DNA and cause the structural damage. But it has

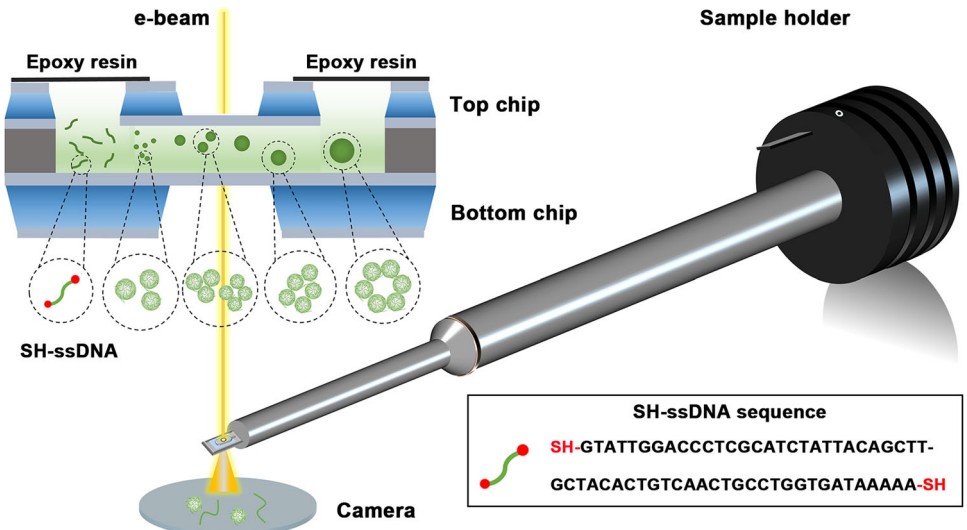

**Fig. 1 LP-TEM depicts the dynamics of DNA.** Schematic showing the LP-TEM observation of disulfide bond-induced aggregation and rearrangement of ssDNA.

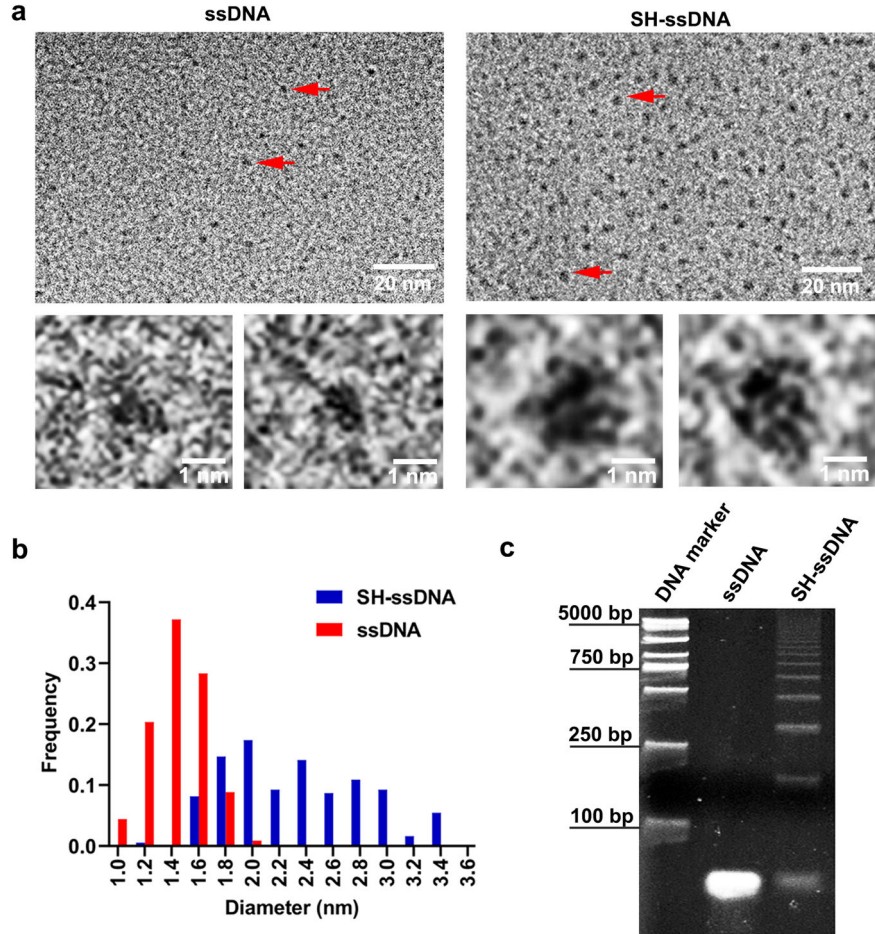

**Fig. 2 Characterization of ssDNA self-assembly induced by the sulfhydryl group. a** TEM images were taken 60 min after ssDNA (left) and SH-ssDNA (light) in the buffer with a 5 μM concentration at dose condition of $60\,e^-\,Å^{-2}s^{-1}$, respectively. Bottom images showed the magnified areas of the red arrows of ssDNA and SH-ssDNA. **b** Size distribution of ssDNA and SH-ssDNA. **c** Image of 5 μM ssDNA and SH-ssDNA macromolecules in liquid after nucleic acid electrophoresis during the 2-h storage at 37 °C.

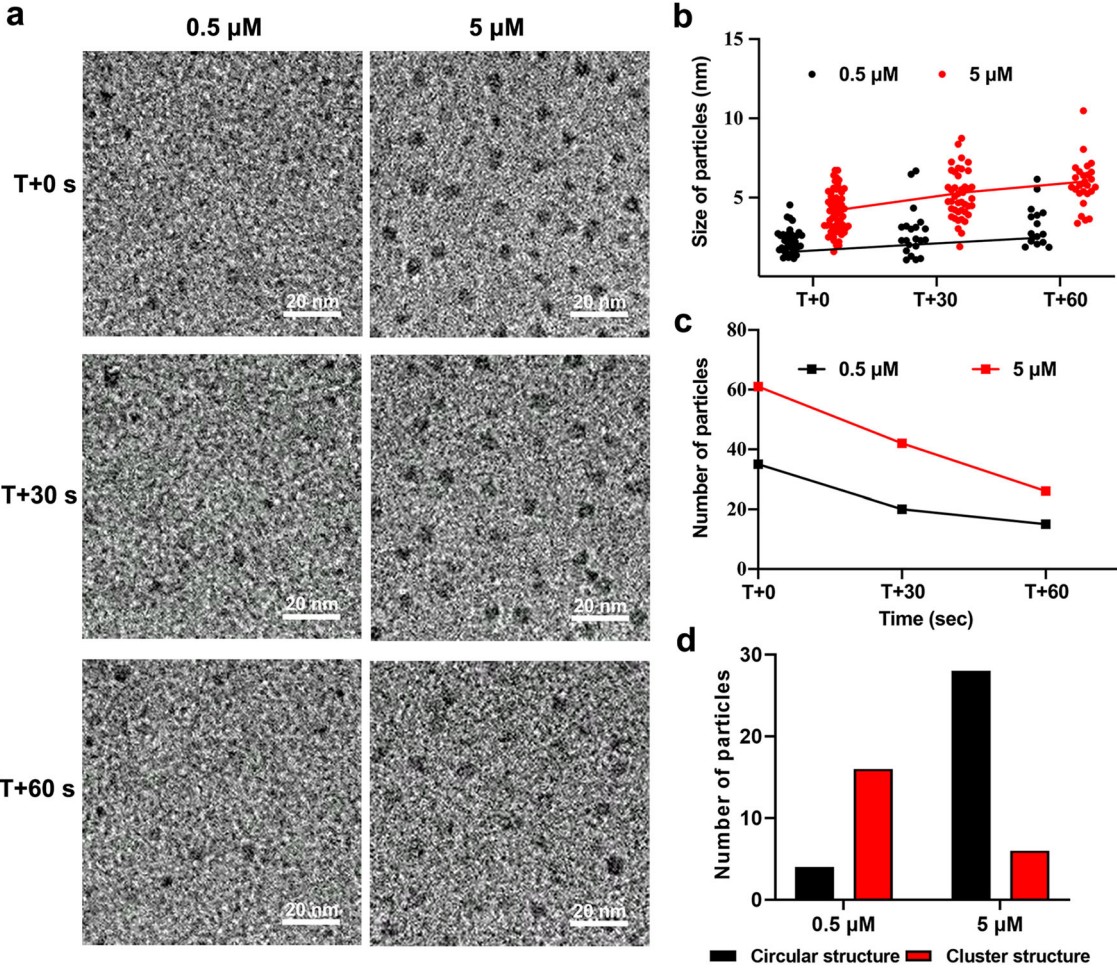

**Fig. 3 LP-TEM observations of SH-ssDNA self-assembly at different imaging times and concentrations. a** Sequential TEM images of 0.5 and 5 μM SH-ssDNA macromolecules in liquid. **b** SH-ssDNA macromolecule diameter as a function of time. **c** Number of SH-ssDNA macromolecules as a function of time. **d** Number of circular and cluster structures of SH-ssDNA at T + 60 s at different concentrations. T is observation start time.

been reported that DNA dynamics was believed to be unaffected when the dose rate is not exceeded $110 \, e^{-1} \, Å^{-2} \, s^{-1}$[19]. On the other hand, the in situ TEM result highly matches with the DNA electrophoresis experiments that avoid interference of electron beam. In addition, the buffer ion pairs in solution might neutralize part of the byproducts of electron radiation water and reduce the impact on DNA molecules.

Surface adsorption may affect the DNA motion. The adsorption on the surface might limit the movement and aggregation, however we still observed that lots of DNA molecule are not fully absorbed on the surface, which move out of focus and become invisible. According to literature[23,27–29], the viscosity of liquid layer close to surface will increase 2–6 order of magnitude, which caused by a layer of ordered liquid near the surface, thus cause much more slower diffusion dynamics than that in the bulk solution. The electrophoresis also clear proof that the SH-ssDNA would aggregate into larger molecules. Therefore, the molecules are close to surface but not fully adsorbed on the surface, and molecules are still be able to move freely and aggregate.

**Disulfide bonds induced macromolecule collisions and rearrangement.** To further explore the characteristic and mechanism of disulfide bond-induced biomolecule assembly and rearrangement, we studied the coalescence process of two SS-cirDNA macromolecules in real time (Fig. 4a and Supplementary Video 5). The observations indicated that SH-ssDNA formed SS-cirDNA through the collision of two macromolecules and the rearrangement of disulfide bond (Fig. 4a–c, Supplementary Fig. 2a–h). During the rearrangement, no obvious disulfide bond openings were observed, and thus, we induced that the structural change occurred before two macromolecules merged (Fig. 4a). Instead, when two macromolecules meet, they were directly transformed into the final structure (Fig. 4a). In addition, one SS-cirDNA macromolecule could merge with a few SS-cirDNA macromolecules simultaneously (Fig. 4d–f, Supplementary Video 6). These results suggest that the disulfide bonds induced coalescence process during macromolecule collisions and generated new SS-cirDNA macromolecules by autonomous disulfide bond exchange, resulting in morphological rearrangement without additional conditions such as a catalyst or nucleophile.

Previous studies have assumed that Brownian motion drives the formation and breakage of disulfide bonds in macromolecules[30], that is, the random collisions of macromolecules could lead to the rearrangement of disulfide bonds. In our study, when the distance between macromolecules was less than 2.8 nm, they would quickly approach with a speed of about 0.64 nm/s, which was higher than the average speed of random motion (Fig. 4a, b and Supplementary Video 5). It is well known that van der Waals force is ubiquitous among biological macromolecules. When the macromolecules approach to few nanometers, the van der Waals force will increase sharply, which would accelerate the macromolecules motion. When the two

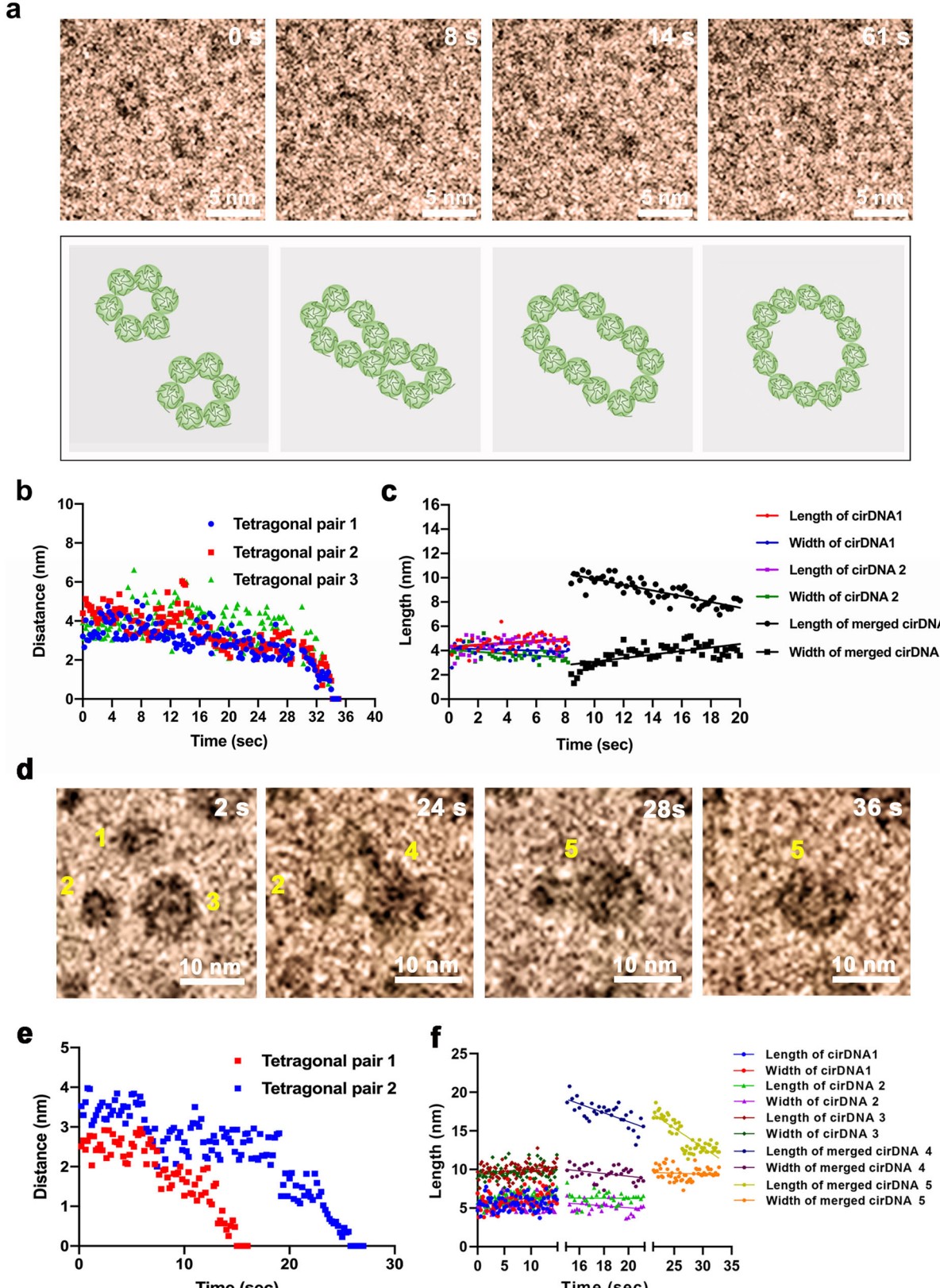

**Fig. 4 Characteristics of SS-cirDNA macromolecule collisions with disulfide bonds. a** Sequential LP-TEM images showing one SS-cirDNA macromolecule formation from two smaller SS-cirDNA macromolecule collisions. **b** Scatter plot showing the distance of two smaller SS-cirDNA macromolecules as a function of time. **c** Scatter plot showing the length and width of two smaller SS-cirDNA macromolecules as a function of time. **d** Sequential LP-TEM images showing one SS-cirDNA macromolecule formation from three SS-cirDNA macromolecule collisions. **e** Scatter plot showing the distance between two smaller SS-cirDNA macromolecules and one lager SS-cirDNA macromolecules as a function of time. **f** Scatter plot showing the length and width of three SS-cirDNA macromolecules as a function of time. T is observation start time.

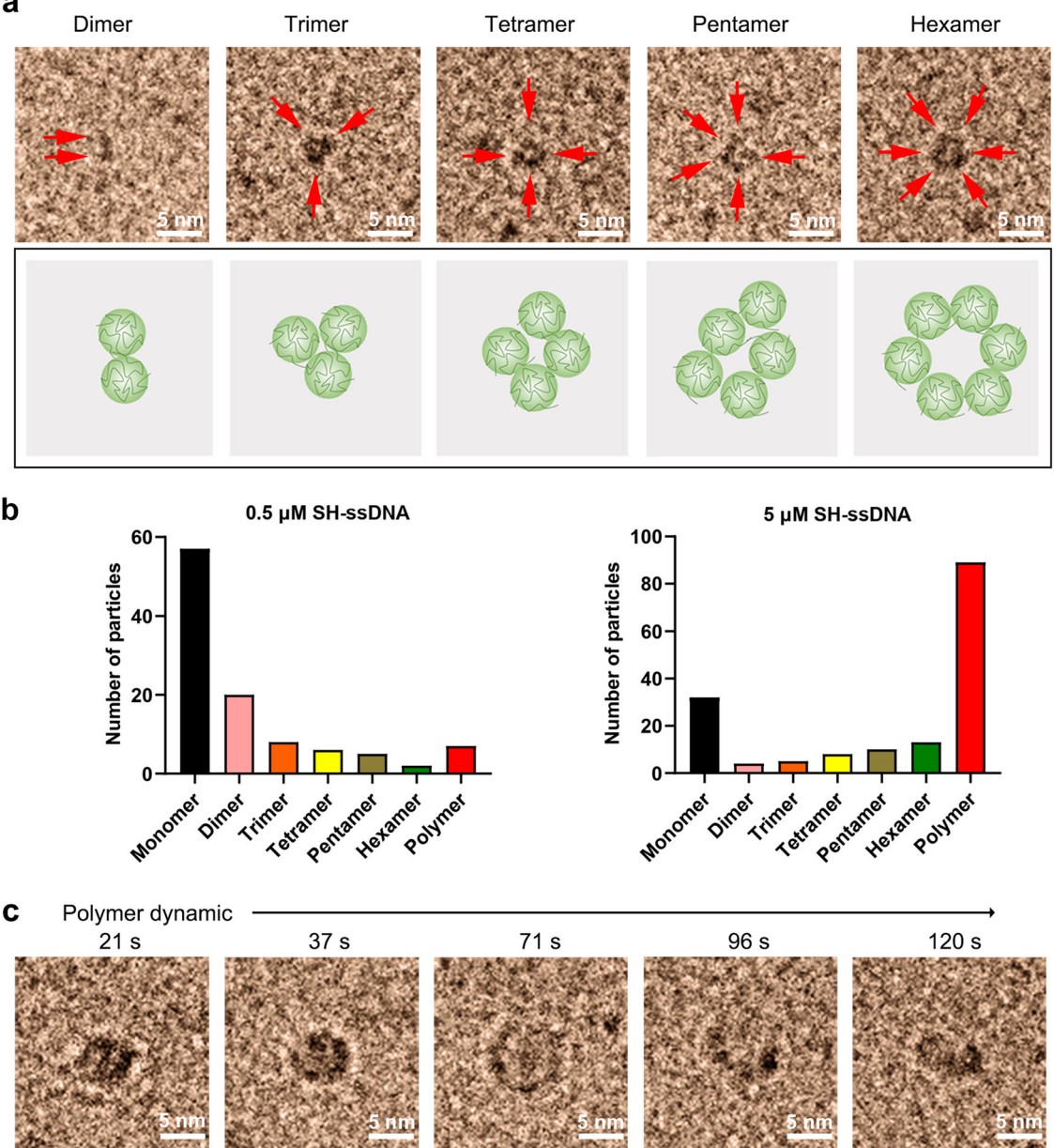

**Fig. 5 Structure and morphology evolution of disulfide bond-modified SH-ssDNA. a** LP-TEM images showing the different forms of SH-ssDNA, single SH-ssDNA as red arrow. **b** Number of the different forms of SH-ssDNA in 0.5 μM and 5 μM SH-ssDNA. **c** Sequential LP-TEM images showing the structural transformation of SS-cirDNA. T is observation start time.

SS-cirDNA approaches, the circular macromolecules reshaped into a wedge shape, which means that macromolecules structure is flexible and the van der Waals force can deform flexible macromolecules (Fig. 4c, f, Supplementary Video 5 and 6).

**Disulfide bond induced SH-ssDNA morphology change.** Several studies have focused on the relationship between disulfide bond formation and macromolecule folding. The hypothesis of disulfide bond formation driving biomolecule folding or vice versa has been proposed in previous studies but without conformity of viewpoint[9]. Hence, we explored the effect of disulfide bonds on the morphological rearrangement of macromolecules. During collision, SS-cirDNA presented significant morphological rearrangement (Fig. 4c and Supplementary Video 5). Owing to the attraction between macromolecules containing the disulfide bond, the macromolecules appeared to rapidly approach, temporary deform, and

non-specifically fuse within a distance of about 2.8 nm, and then exhibited flexible macromolecule rearrangement and morphological change within a short period of about 20 s. These temporary structure deformations may explain why thioredoxin non-specifically integrates with substrates and catalyzes disulfide bond rearrangement of the substrate.

In addition, disulfide bond-modified SS-cirDNA presented different forms of macromolecules, including dumbbell-shaped dimers, triangular trimers, tetrahedral tetramers, and flexible, circular shaped polymers (Fig. 5a, b and Supplementary Video 7). SH Group is the driving force of molecule aggregation, but as showed in Fig. 3b, different forms are related to concentration, if the concentration is high, concentrated into large particles; when the concentration is low, there are more small particles. Moreover, these SH-ssDNA macromolecules, especially polymers, undergo frequent morphology changes with free rolling and

twisting, which shows the real-time morphology of nano-ring biomacromolecules in solution (Fig. 3b). These results indicate that disulfide bonds significantly alter the structure and size of macromolecules, providing a new perspective on the function of sulfhydryl macromolecules.

## Conclusion

;In summary, LP-TEM enables the direct capture of the dynamic behavior of ssDNA and SH-ssDNA in the native environment. In this study, we observed that the sulfhydryl groups induced ssDNA molecules to self-assemble and rearrange at nanometer resolution in real time. In addition, the disulfide bond interaction triggered the aggregation of two SS-cirDNA macromolecules along with significant structural changes, which have been neither revealed nor described earlier, they can help us to understand the formation and rearrangement of disulfide bonds in biomolecules. Moreover, the LP-TEM, as a visualization tool, holds great potential in biomedicine, especially in the high spatial observation of smaller macromolecules, including the structural changes and interactions of proteins.

## Materials and methods

**Preparation of biomacromolecules**. SH-ssDNA and ssDNA were purchased from Sangon Biotech (China), and the sequences are shown as follow. Sequence of SH-ssDNA: SH-GTATTGGACCC

TCGCATCTATTACAGCTTGCTACACTGTCAACTGCCTGGTGA-
TAAAAA-SH, Sequence of ssDNA:
GTATTGGACCCTCGCATCTATTACAGCTTGCTACACTG
TCAACTGCCTGGTGATAAAAA.

The SH-ssDNA stock solution was prepared by adding 100 μL Phosphate Buffer Saline (PBS) into a powder of 0.5 optical density SH-ssDNA, and the 0.5 or 5 μM SH-ssDNA final solution was prepared by diluting the stock solution with PBS. The ssDNA final solutions were also prepared as in the same manner. After 30 min standing at room temperature, the samples were loaded for imaging.

**Liquid cell preparation and in situ TEM imaging process**. A static liquid cell with 10-nm low-stress silicon nitride membranes on silicon wafers was purchased from Chip-nova (Xiamen, China) and the fabrication processes of liquid cell we used here follows the method of the previous literature[13,14]. 10 nm silicon nitride is selected as the window membrane to reduce the background noise. Silicon nitride surface is also inert and the liquid cell with 10 nm silicon nitride membrane also have many applications in the field of in situ TEM imaging of biological samples. The spacer between top chip and bottom chip is about 100 nm, then, a small amount (~100 nL) of the solution with DNA was injected into one of the cavities in liquid cell. The solution was sucked into the cell by capillary force and formed a thin liquid layer (~100 nm) sandwiched between two silicon nitride membranes. After the sample liquid was loaded into the liquid cell through the injection hole, the hole was sealed by epoxy. Properly sealing the liquid cell maintains the liquid inside the liquid cell for an extended period, which is critical for enabling DNA molecule interaction and macromolecule formation.

All videos were recorded by using TVIPS high sensitivity CMOS camera, which allows imaging at dose condition of 60 e⁻·Å⁻²s⁻¹. Images in Supplementary Videos 1–7 were obtained under TECNAI-F20(200 Kv), Talos F200s transmission electron microscope. Each video plays 6 times faster than real time (i.e., one frame per second). Biomacromolecules in the sequential images in Figs. 2–5 were processed by automatic toning and analysis by Image J. Seven original images can be retrieved. See Supplementary Videos 1–7.

**Nucleic acid electrophoresis experiments**. The prepared samples of SH-ssDNA and ssDNA were incubated for 24 h at 37 °C, and then diluted with 10 μL loading buffer and detected by 8% denatured polyacrylamide gel electrophoresis (110 V, 110 min). Samples were then treated by GelStain for observation by an automatic gel imaging analyzer (JS-780, Shanghai). Each experiment was performed in triplicate.

**Reporting summary**. Further information on research design is available in the Nature Portfolio Reporting Summary linked to this article.

## Data availability

All relevant data are included in this manuscript and supplementary data. The individual video files are available in Supplementary Videos 1–7.

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

## Acknowledgements

This research was funded by the National Natural Science Foundation of China (Grant 32101217, 3210120393, 22288102, 21991151, 21991150, 91934303) and the National Key Research and Development Program of China (No. 2017YFA0206500). We also thank TopEdit (www.topeditsci.com) for linguistic assistance during manuscript preparation.

## Author contributions

F.W.Z. and Y.H.J. designed and synthesized the biomacromolecules. F.W.Z., Y.H.J., N.N.H., D.W., Y.L., G.W.C., J.F.C. and H.G.L. designed the experiments. F.W.Z., Y.H.J., T.Q.Z., M.Q. and Y.S. performed the LP-TEM experiments. F.W.Z., T.T.G. and S.T.C. performed the nucleic acid electrophoresis experiments. F.W.Z., S.G.S. and H.G.L. wrote the paper. All authors contributed to the experimental design, data analysis, and discussions.

## Competing interests

The authors declare no competing interests.
