## [Peer Review File · Communications Chemistry]

Real-time imaging of sulfhydryl single-stranded DNA aggregationReviewers' comments:

Reviewer #1 (Remarks to the Author):

The authors have done an excellent job imaging SS-DNA strand dynamics using in liquid EM. They are to be congratulated on this feat alone.

I do think this paper should be published, however, there are some issues with interpretation as a simple S-S bond formation or exchange. There are very few controls to isolate this bond and the spatial resolution is no way good enough to state that the chemical action is occurring solely at the S-H site as a unique bond activation process. The authors do not seem to appreciate the harsh environment that the shower of probe electrons produce though the very scattering process exploited for imaging.

The authors should consider:

- state explicitly the electron dose and give this key factor such a short shrift in their treatment. They need to educate themselves better on the electron pulse radiolysis processes going on in parallel. See work of F. Ross, Sneider on this essential process. Even if the authors want to insist "low dose" there will still be electron radical chemistry induced by the electron beam and the time window is well sufficient for even minute channels to lead to an effect.

- the authors have to consider other strand breaks, radical processes than just S-S bond processes and exchanges. Radicals, bond breaking with radical formation will ALSO form cross linkage to other DNA strands.

- the only distinction comparing SH-ssDNA is the video of ssDNA clustering dynamics. There is not much quantitative done with this one control and it is not convincing as the SH groups will change the surface adhesion and prospect for collision induced growth..with nothing to do with S-S exchange processes.

- The authors state ssDNA forms a double helix. This would be unusual and not sure this is what they meant and how they determined this point.

TO be concise, there resolution is no way good enough to uniquely assign the circulization of DNA and formation of clusters solely to S-S bond exchange. The authors need to consider the multitude of other pathways and properly cite other possible channels. The resolution should be significantly improved if they wish to make any claims that the chemistry is occurring at the SH points. Also they need to respond to how they could even observe the DNA circular clusters as simple estimates of translational diffusion and especially rotational diffusion should wash out details. It is clear that surface effects are retarding the DNA motion and therefore surface effects will also affect the chemistry. The SH groups may be doing nothing more than increasing the surface adsorption and leading to better resolution of clusters that stick more.

I hope the authors take the above comments as constructive. These are beautiful results. They just need to be more open to other interpretations than a single point chemistry at SH points. Certainly better resolution and controls (post analysis) is needed to claim S-S, SH specific chemistry, nevermind trying to make the link to role of disulphide bond exchanges to protein structures. I am sure with some more circumspect discussion of possible mechanisms this paper should be published. It is an important step that follows up on previous work (refs 19-21) with better resolution.

Reviewer #2 (Remarks to the Author):

The manuscript by Zeng claimed to report real time imaging of aggregation and rearrangement of

single-stranded DNA as triggered by disulfide bond chemistry. Authors started to present images that matched up with the size of single-stranded DNA, whose size were observed to increase for a sample solution contains higher concentration of single-stranded DNA. The larger size objects were interpreted as aggregates modulated by disulfide bond chemistry. Authors identified some coalescence events and presented how smaller objects evolve towards larger objects. Lastly, authors picked some images as diverse forms of aggregates, from dimer to trimer, to hexamer, and some morphological changes of one particle as a function of time. Overall, not enough control experiments and quantitative analysis were performed to support the claim. Factors that are known to affect chemistry and motion of these molecular processes such as surface adsorption and electron beam induced effects were not mentioned and discussed. I would not recommend publication in communication chemistry at this stage, it may worth reconsideration after major revisions in addressing issues about reproducibility and over-interpretations of data.

Some general comments as follows.

1. The title does not convey accurately about what was observed. Authors observed aggregation of single-stranded DNA and the rearrangement of aggregates, not rearrangement of a single molecule, now it reads confusing.
2. According to data presented in figures, I am only convinced by the observation of individual molecules and formation of aggregates presented in Figure 2 and 3, respectively. Proper quantification and reasoning are needed for the detailed interpretation about structure and chemistry, rather than by imagination and guess. For example: (1) how authors identify close and open loop from the present image quality in Figure 2; From line 148-154, authors ascribe open-looped structure to free sulfhydryl, how do they rule out the possibility that these structures could also be bent closed-loop structures. It would be more convincing to have the statistics of width or size for all open-looped and closed-looped structure; (2) in line 99 and Fig. 2C, authors claimed to resolve the helical structure, however, helix is not an expected structure for the base sequence of ssDNA that the authors used, usually it refers to dsDNA and it has a fixed pitch size; (3) there is no statistics about the reproducibility and representativeness for images and processes presented in Figure 4 and 5; (4) authors failed to explain how they ascribe images to different forms of aggregates, dimer, trimer, tetramer, pentamer, and hexamer, and how and why they should form in Figure 5.
3. Discussions of the known electron beam induced effects are lacking. Heating, charging¹ and radiolysis², these effects are in particular important for damage-prone biological samples, unlike nanoparticles. Although experiments from other groups have proven the possibility to image biomacromolecules, for example peptide assembly³, experimental conditions can be very different in experiments reported here. For example, graphene liquid cell is known to better protect fragile sample from radiation damage⁴ and the surface is more inert. Even for the similar silicon nitride window, as authors used thinner membrane, gap distance, electron dose, chips from different manufactures, more careful evaluation needs to be done.
4. Authors included SH-RNA as samples, however it does not seem to integrate with the main body of research.
5. Some interpretations of motion are not proper, such as the discussion of Brownian motion in lines of 173 to 183. Authors should have noted that surface mediated long jumps can occur for Brownian motion, which can lead to seemingly larger displacements for some steps. The surface effect could be very complicated in a silicon nitride liquid cell for sticking prone biomacromolecules and can be system specific, perhaps, authors should at least show that mean square displacements (MSD) with a slope 1 for a control ssDNA. With this verification, in order to understand whether the attraction indeed exceeds the range of Brownian motion, authors should then analyze the trajectory and MSD with some reproducible statistics of both SH-ssDNA and the control ssDNA.

Some specific comments as follows.

1. Abbreviation should be explained when it first appears, like ssDNA.
2. In line 113, mean square displacement (MSD) of the ssDNA center of mass will better quantify the movement of ssDNA to support your claim of "Brownian motion was significantly restricted".
3. In line 116, it is not convincing to connect macromolecules size to the conclusion that a disulfide bond form by active dehydrogenation without any other experimental evidence.
4. From line 132 to 143, this paragraph is only a description of the experiment without corresponding analysis or discussion:
In line 132, the authors claim that "Interactions between macromolecules drive spontaneous and continuous self-assembly of SH-ssDNA" without specifying weather the interaction they referred is related to disulfide bond.
In line 136, the authors mention that "SH-ssDNA molecules presented different states at different concentrations" without any explanation or discussion.
5. In line 161, "no obvious disulfide bond opening was observed" is not supported by images presented at such a limited resolution. Thus, the conclusion that "SS-cirDNA rearranges through a disulfide-disulfide exchange path instead of a thiol-disulfide exchange path" is not well-supported unless further experimental evidence is provided.
6. In line 184, the title should be changed into "Disulfide bond induced HS-ssDNA morphology change"
7. In line 259, the total amount of data involved in the statistical process should be indicated.

References:

- (1) Grogan, J. M.; Schneider, N. M.; Ross, F. M.; Bau, H. H. Bubble and pattern formation in liquid induced by an electron beam. *Nano Lett.* 2014, 14 (1), 359-364.
- (2) Korpanty, J.; Parent, L. R.; Gianneschi, N. C. Enhancing and mitigating radiolytic damage to soft matter in aqueous phase liquid-cell transmission electron microscopy in the presence of gold nanoparticle sensitizers or isopropanol scavengers. *Nano Lett.* 2021, 21 (2), 1141-1149.
- (3) Touve, M. A.; Carlini, A. S.; Gianneschi, N. C. Self-assembling peptides imaged by correlated liquid cell transmission electron microscopy and MALDI-imaging mass spectrometry. *Nat. Commun.* 2019, 10 (1), 4837.
- (4) Cho, H.; Jones, M. R.; Nguyen, S. C.; Hauwiller, M. R.; Zettl, A.; Alivisatos, A. P. The use of graphene and Its derivatives for liquid-phase transmission electron microscopy of radiation-sensitive specimens. *Nano Lett.* 2017, 17 (1), 414-420. Keskin, S.; de Jonge, N. Reduced radiation damage in transmission electron microscopy of proteins in graphene liquid cells. *Nano Lett.* 2018, 18 (12), 7435-7440.

Response to reviewers

Reviewer #1 (Remarks to the Author):

The authors have done an excellent job imaging SS-DNA strand dynamics using in liquid EM. They are to be congratulated on this feat alone.

I do think this paper should be published, however, there are some issues with interpretation as a simple S-S bond formation or exchange.

There are very few controls to isolate this bond and the spatial resolution is no way good enough to state that the chemical action is occurring solely at the S-H site as a unique bond activation process. The authors do not seem to appreciate the harsh environment that the shower of probe electrons produce though the very scattering process exploited for imaging.

The authors should consider:

- state explicitly the electron dose and give this key factor such a short shrift in their treatment. They need to educate themselves better on the electron pulse radiolysis processes going on in parallel. See work of F. Ross, Sneider on this essential process. Even if the authors want to insist "low dose" there will still be electron radical chemistry induced by the electron beam and the time window is well sufficient for even minute channels to lead to an effect.
- the authors have to consider other strand breaks, radical processes than just S-S bond processes and exchanges. Radicals, bond breaking with radical formation will ALSO form cross linkage to other DNA strands.

Response to reviewer: Thanks for the comment. The possible damage caused by the electron beam have to be considered during the in situ liquid phase TEM test. Water molecule radiolysis by the electrons would generate few species, including hydrated (solvated) electrons e_{aq}^- , hydrogen radical, $H\bullet$, hydroxyl radical $OH\bullet$, and H_2 . In our experiment, the electron dose is $60 e \text{ \AA}^{-2}s^{-1}$. According to the reference [1], the concentration of the above species is around 10^{-7} to 10^{-5} mol/L which is a fairly low concentration and the influence on the reaction between SH-DNA is negligible. Parallel electron beams interact with DNA, which can disrupt DNA dynamics and cause structural damage. However, it has been reported that DNA dynamics was believed to be unaffected when the dose rate is not exceeded $110 e \text{ \AA}^{-2}s^{-1}$. [2] On the other hand, the in situ TEM result highly matches with the DNA electrophoresis experiments that without interference of electron beam. In addition, the buffer ion pairs in solution might neutralize part of the byproducts of electron radiation water and reduce the impact on DNA molecules.

[1] Schneider N M, Norton M M, Mendel B J, et al. Electron–water interactions and implications for liquid cell electron microscopy[J]. The Journal of Physical Chemistry C, 2014, 118(38): 22373-22382.

[2] S. Keskin, S. Besztejani, G. Kassier, S. Manz, R. Bückner, S. Riekeberg, H. K. Trieu, A. Rentmeister, R. J. D. Miller, Visualization of multimerization and self-assembly of DNA-functionalized gold nanoparticles using in-liquid transmission electron microscopy. J. Phys. Chem. Lett. 6, 4487-4492 (2015).

- the only distinction comparing SH-ssDNA is the video of ssDNA clustering dynamics. There is not much quantitative done with this one control and it is not convincing as the SH groups will change the surface adhesion and prospect for collision induced growth with nothing to do with S-S exchange processes.

Response to reviewer : We appreciate the reviewer's suggestion. Considering that in the liquid cell environment, SH groups may alter surface adhesion, and then affects molecular agglomeration. We

performed aggregation experiments and electrophoresis in bulk solution, and we still found that SH-ssDNA had obvious aggregation compared with ssDNA (Fig 2c).

The SH group may change the surface adhesion, but we found significant movement of the molecules with 2.6 nm/s (Fig 4ab), and some movie shows the molecules move out of focus and cannot be seen in the field of view, which indicate they are not fully adsorbed on surface.

Our current resolution is indeed insufficient to observe detail S-S bond exchange with SH sites, so we deleted the content of S-S exchange in the manuscript. We will try to improve the resolution in the future study to obtain more clear and solid results.

- The authors state ssDNA forms a double helix. This would be unusual and not sure this is what they meant and how they determined this point.

Response to reviewer: Thanks reviewer for pointing it out, due the hydrophobic force of ssDNA, these ssDNAs in solution will shrink to a dot, and the SH-ssDNA shrink to a string-like structure due to the SH-would aggregate to form S-S bond, the word helix indeed cannot present the shape changes, we change it according in the main text.

TO be concise, there resolution is no way good enough to uniquely assign the circularization of DNA and formation of clusters solely to S-S bond exchange. The authors need to consider the multitude of other pathways and properly cite other possible channels. The resolution should be significantly improved if they wish to make any claims that the chemistry is occurring at the SH points.

Response to reviewer : We appreciate the reviewer's suggestion. Our current resolution is indeed insufficient to observe detail of bonds, so we changed the description accordingly. We will improve our experiment with better imaging techniques and instruments, hope we could identify detail structure changes with higher resolution in the future.

Also they need to respond to how could even observe the DNA circular clusters as simple estimates of translational diffusion and especially rotational diffusion should wash out details.

Response to reviewer : Thanks reviewer for pointing it out. Figure A shows translational diffusion, we can see that the molecules did not move out focus plane, and the positions of the circular molecules do not change significantly on the Z-axis. If the molecules move up and down in solution, we could not image them all the

time, some movie shows the molecules move out of focus and cannot continue to see them. There is a typical rotational diffusion in Figure 5c, where we can see that the white edge appears at 37s and disappears at 120s.

It is clear that surface effects are retarding the DNA motion and therefore surface effects will also affect the chemistry. The SH groups may be doing nothing more than increasing the surface adsorption and leading to better resolution of clusters that stick more.

Response to reviewer : Surface adsorption may affect the DNA motion. The adsorption on the surface might limit the movement and aggregation, however we still observed that lots of DNA molecule are not fully adsorbed on the surface, which move out of focus and become invisible. According to literature [1], the viscosity of liquid layer close to surface will increase 2 orders of magnitude, thus caused much slower diffusion dynamics than that in bulk solution. The electrophoresis also clear proof that the SH-ssDNA would aggregate into larger molecules. Therefore, we think the molecule are close to surface but not fully adsorbed on the surface, and molecules are still be able to move freely and aggregate.

[1] Powers, A.S., H.G. Liao, S.N. Raja, N.D. Bronstein, A.P. Alivisatos, and H. Zheng, Tracking Nanoparticle Diffusion and Interaction during Self-Assembly in a Liquid Cell. *Nano Lett*, 2017. 17(1): p. 15-20.

I hope the authors take the above comments as constructive. These are beautiful results. They just need to be more open to other interpretations than a single point chemistry at SH points. Certainly better resolution and controls (post analysis) is needed to claim S-S, SH specific chemistry, nevermind trying to make the link to role of disulphide bond exchanges to protein structures. I am sure with some more circumspect discussion of possible mechanisms this paper should be published. It is an important step that follows up on previous work (refs 19-21) with better resolution.

Response to reviewer: Thank you for your kind suggestion. Our analysis and understanding need to be

strengthened. We will continue to study this topic in depth and improve imaging resolution, thus obtain solid results which could reveal the detail interaction mechanism of DNA molecules.

Reviewer #2 (Remarks to the Author):

The manuscript by Zeng claimed to report real time imaging of aggregation and rearrangement of single-stranded DNA as triggered by disulfide bond chemistry. Authors started to present images that matched up with the size of single-stranded DNA, whose size were observed to increase for a sample solution contains higher concentration of single-stranded DNA. The larger size objects were interpreted as aggregates modulated by disulfide bond chemistry. Authors identified some coalescence events and presented how smaller objects evolve towards larger objects. Lastly, authors picked some images as diverse forms of aggregates, from dimer to trimer, to hexamer, and some morphological changes of one particle as a function of time.

Overall, not enough control experiments and quantitative analysis were performed to support the claim.

Factors that are known to affect chemistry and motion of these molecular processes such as surface adsorption and electron beam induced effects were not mentioned and discussed. I would not recommend publication in communication chemistry at this stage, it may worth reconsideration after major revisions in addressing issues about reproducibility and over-interpretations of data.

Some general comments as follows.

1. The title does not convey accurately about what was observed. Authors observed aggregation of single-stranded DNA and the rearrangement of aggregates, not rearrangement of a single molecule, now it reads confusing.

Response to reviewer: Thanks reviewer for pointing it out. We revised the title as follows: Real-time Imaging of Disulfide Bond induced Aggregation of Single-stranded DNA.

2. According to data presented in figures, I am only convinced by the observation of individual molecules and formation of aggregates presented in Figure 2 and 3, respectively. Proper quantification and reasoning are needed for the detailed interpretation about structure and chemistry, rather than by imagination and guess. For example:

(1) how authors identify close and open loop from the present image quality in Figure 2; From line 148-154, authors ascribe open-looped structure to free sulfhydryl, how do they rule out the possibility that these structures could also be bent closed-loop structures. It would be more convincing to have the statistics of width or size for all open-looped and closed-looped structure;

Response to reviewer : Thanks reviewer for pointing it out. Our current resolution is indeed insufficient to distinguish of closed and opened loop, so we deleted the content of close and open loop in the manuscript. We re-counted the particles and divided them into two groups: the circular group (with middle gaps) and the cluster group (no voids in the center), as Figure 3d.

(2) in line 99 and Fig. 2C, authors claimed to resolve the helical structure, however, helix is not an expected structure for the base sequence of ssDNA that the authors used, usually it refers to dsDNA and it has a fixed pitch size;

Response to reviewer : Thanks reviewer for pointing it out. We have edited it. We use “string-like structure” to described the shape in Fig. 2a.

(3) there is no statistics about the reproducibility and representativeness for images and processes presented in Figure 4 and 5;

Response to reviewer: We appreciate the reviewer's suggestion. We have added 8 pairs quantitative analysis of macromolecules aggregation in the Supplementary Data for Figure 4, and quantitative analysis of diverse forms of aggregates in Figure 5b, which can provide data to verify repeatability. We have revised the content in the manuscript.

(4) authors failed to explain how they ascribe images to different forms of aggregates, dimer, trimer, tetramer, pentamer, and hexamer, and how and why they should form in Figure 5.

Response to reviewer : Thanks reviewer for pointing it out. We classify the different polymers according to their morphology (Fig. 5a), as shown by red arrows. We also counted the number of different polymers in 0.5 μM and 5 μM SH-ssDNA, as shown in Fig. 5b.

3. Discussions of the known electron beam induced effects are lacking. Heating, charging¹ and radiolysis², these effects are in particular important for damage-prone biological samples, unlike nanoparticles. Although experiments from other groups have proven the possibility to image biomacromolecules, for example peptide assembly³, experimental conditions can be very different in experiments reported here. For example, graphene liquid cell is known to better protect fragile sample from radiation damage⁴ and the surface is more inert. Even for the similar silicon nitride window, as authors used thinner membrane, gap distance, electron dose, chips from different manufactures, more careful evaluation needs to be done.

Response to reviewer : We appreciate the reviewer's comments. The possible damage caused by the electron beam have to be considered during the in-situ liquid phase TEM test. Water molecule radiolysis by the electrons would generate few species, including hydrated (solvated) electrons e_{aq}^- , hydrogen radical, $H\cdot$, hydroxyl radical $OH\cdot$, and H_2 . In our experiment, the electron dose is $60 e \text{ \AA}^{-2} s^{-1}$. According to the reference [1], the concentration of the above species is around 10^{-7} to 10^{-5} mol/L, which is a fairly low concentration and the effect on the reaction between SH-DNA is negligible. The parallel electron beam interacts with DNA and this might disturb the dynamics of DNA and cause the structural damage. But it has been reported that DNA dynamics was believed to be unaffected when the dose rate is not exceeding $110 e \text{ \AA}^{-2} s^{-1}$ [2]. On the other hand, the in situ TEM result highly matches with the DNA electrophoresis experiments that without interference of electron beam. In addition, the buffer ion pairs in solution might neutralize part of the byproducts of electron radiation water and reduce the impact on DNA molecules.

Surface adsorption may affect the DNA motion. The adsorption on the surface might limit the movement and aggregation, however we still observed that lots of DNA molecule are not fully adsorbed on the surface, which move out of focus and become invisible. According to literature [3], the viscosity of liquid layer close to surface will increase 2 order of magnitude, thus caused much more slower diffusion dynamics. The electrophoresis also clear proof that the SH-ssDNA would aggregate into larger molecules. Therefore, we think the molecule are close to surface but not fully adsorbed on the surface, and molecules are still be able to move freely and aggregate. These discussions are added to manuscript accordingly.

[1] Schneider N M, Norton M M, Mendel B J, et al. Electron–water interactions and implications for liquid cell electron microscopy[J]. The Journal of Physical Chemistry C, 2014, 118(38): 22373-22382.

[2] S. Keskin, S. Besztejan, G. Kassier, S. Manz, R. Bücken, S. Riekeberg, H. K. Trieu, A. Rentmeister, R. J. D. Miller, Visualization of multimerization and selfassembly of DNA-functionalized gold nanoparticles using in-liquid transmission electron microscopy. J. Phys. Chem. Lett. 6, 4487-4492 (2015).

[3] Powers, A.S., H.G. Liao, S.N. Raja, N.D. Bronstein, A.P. Alivisatos, and H. Zheng, Tracking Nanoparticle Diffusion and Interaction during Self-Assembly in a Liquid Cell. Nano Lett, 2017. 17(1): p. 15-20.

4. Authors included SH-RNA as samples, however it does not seem to integrate with the main body of research.

Response to reviewer : Thanks reviewer for pointing it out, SH-RNA indeed didn't integrate with the main body of research, so we deleted the content of SH-RNA in the manuscript.

5. Some interpretations of motion are not proper, such as the discussion of Brownian motion in lines of 173 to 183. Authors should have noted that surface mediated long jumps can occur for Brownian motion, which can lead to seemingly larger displacements for some steps. The surface effect could be very complicated in a silicon nitride liquid cell for sticking prone biomacromolecules and can be system specific, perhaps, authors should at least show that mean square displacements (MSD) with a slope 1 for a control ssDNA. With this verification, in order to understand whether the attraction indeed exceeds the range of Brownian motion, authors should then analyze the trajectory and MSD with some reproducible statistics of both SH-ssDNA and the control ssDNA.

Response to reviewer : Thanks reviewer' s comment. Indeed, the interpretations of motion here was not proper, we have added the result of mean square displacement and changed the description in manuscript.

Some specific comments as follows.

1. Abbreviation should be explained when it first appears, like ssDNA.

Response to reviewer : Thanks reviewer for pointing it out. We have change it in manuscript.

2. In line 113, mean square displacement (MSD) of the ssDNA center of mass will better quantify the movement of ssDNA to support your claim of “Brownian motion was significantly restricted”.

Response to reviewer : Thanks reviewer for pointing it out. Indeed, the Brownian motion was not restricted, we have added the result of mean square displacement showed in Supplementary Figure 1, the MSD is $11.06 \text{ nm}^2/\text{s}$ $5\mu\text{M}$ SH-ssDNA aggregates, they show a typical characteristic of Brownian motion according to literature (1).

(1) Andreas Verch 1, Marina Pfaff 1, Niels de Jonge. Exceptionally Slow Movement of Gold Nanoparticles at a Solid/Liquid Interface Investigated by Scanning Transmission Electron Microscopy. *Langmuir*. 2015 Jun 30;31(25):6956-64.

3. In line 116, it is not convincing to connect macromolecules size to the conclusion that a disulfide bond form by active dehydrogenation without any other experimental evidence.

Response to reviewer : Thanks reviewer for pointing it out. Given the two SH group merged a disulfide bond (-S-S-), we speculated that the process contains dehydrogenation, but we did not observe this effect experimentally. Therefore, we have deleted the sentence of "by active dehydrogenation".

4. From line 132 to 143, this paragraph is only a description of the experiment without corresponding analysis or discussion:

In line 132, the authors claim that “Interactions between macromolecules drive spontaneous and continuous self-assembly of SH-ssDNA” without specifying whether the interaction they referred is related to disulfide bond.

In line 136, the authors mention that “SH-ssDNA molecules presented different states at different concentrations” without any explanation or discussion.

Response to reviewer : Thanks reviewer for pointing it out. Considering that we have already mentioned "SH-ssDNA self-curved into circular shaped SS-cirDNA" in lines 104-106, we deleted this sentence of “Interactions between macromolecules drive spontaneous and continuous self-assembly of SH-ssDNA” in line 132.

We have added corresponding analysis in Figure 3d and discussion in 128-136, showed as follow.

“We further observed that the larger particles showed circular structure with middle gaps, while the smaller particles showed cluster structure with no voids in the center. According to the statistics of circular particles and cluster particles respectively, it was found that compared with 0.5 μM SH-ssDNA, there were more circular particles in the group of 5 μM SH-ssDNA (Fig. 3d), which might be due to the increase of concentration, and then increase the possibility of particle collision and aggregation. These results showed assemble behavior of SH-ssDNA is concentration dependent”.

5. In line 161, “no obvious disulfide bond opening was observed” is not supported by images presented at such a limited resolution. Thus, the conclusion that “SS-cirDNA rearranges through a disulfide–disulfide exchange path instead of a thiol-disulfide exchange path” is not well-supported unless further experimental evidence is provided.

Response to reviewer : Thanks reviewer for pointing it out, our current resolution is indeed insufficient to observe detail S-S bond exchange and opened disulfide bond, so we deleted the content of S-S exchange in the manuscript. We will try to improve the resolution in the future study and obtain more clear results.

6. In line 184, the title should be changed into “Disulfide bond induced SH-ssDNA morphology change”

Response to reviewer : Thanks reviewer for pointing it out. We have changed it in the manuscript.

7. In line 259, the total amount of data involved in the statistical process should be indicated.

Response to reviewer : Thanks reviewer for pointing it out. We performed statistical analysis in the result of RNA Nucleic acid electrophoresis experiments, considering that SH-RNA indeed didn't integrate with the main body of research, so we deleted the content of SH-RNA and related statistical process in the manuscript.

References:

- (1) Grogan, J. M.; Schneider, N. M.; Ross, F. M.; Bau, H. H. Bubble and pattern formation in liquid induced by an electron beam. *Nano Lett.* 2014, 14 (1), 359-364.
- (2) Korpanty, J.; Parent, L. R.; Gianneschi, N. C. Enhancing and mitigating radiolytic damage to soft matter in aqueous phase liquid-cell transmission electron microscopy in the presence of gold nanoparticle sensitizers or isopropanol scavengers. *Nano Lett.* 2021, 21 (2), 1141-1149.
- (3) Touve, M. A.; Carlini, A. S.; Gianneschi, N. C. Self-assembling peptides imaged by correlated liquid cell transmission electron microscopy and MALDI-imaging mass spectrometry. *Nat. Commun.* 2019, 10 (1), 4837.
- (4) Cho, H.; Jones, M. R.; Nguyen, S. C.; Hauwiller, M. R.; Zettl, A.; Alivisatos, A. P. The use of graphene and Its derivatives for liquid-phase transmission electron microscopy of radiation-sensitive specimens. *Nano Lett.* 2017, 17 (1), 414-420. Keskin, S.; de Jonge, N. Reduced radiation damage in transmission electron microscopy of proteins in graphene liquid cells. *Nano Lett.* 2018, 18 (12), 7435-7440.

REVIEWERS' COMMENTS:

Reviewer #1 (Remarks to the Author):

I appreciate the effort made to reply to my comments where my major concerns were related to the lack of adequate discussion of electron induced radical chemistry that would lead to multiple aggregation points not just at disulphide bonds. This point is covered. However, there is still a problem with the lack of appreciation of surface effects. In particular the statement "According to literature [25], the viscosity of liquid layer close to surface will increase 2 order of magnitude, thus caused much more slower diffusion dynamics than that in the bulk solution."

This statement is wrong. The cited paper is incorrect on this point or at least the authors should be more critical. How could the viscosity be that much higher? There is a slight propensity out to 3 waters in forming higher degree of H bonding of water at graphene surfaces...but not out many nm. Surface adhesion will give an effective increase in what would look like slow diffusive motion - even in considering the mean square displacement as a function of time. There only have to be enough binding sites close enough to look diffusive in the spatial and temporal time resolution. This discussion must be tempered in which the role of the surface is given equal weight.

Final comment, the title needs to be changed as I still don't buy their argument on the disulphide bonds being the only source for aggregation.

Please change "Real-time Imaging of Disulfide Bond induced Aggregation of Single-stranded DNA" to "Real-time Imaging of Single-Strand DNA Aggregation".

The authors should state throughout the manuscript that the observations are consistent/correlated with disulphide bond cleavage and aggregation but there are also other possible radiolysis processes that likely also contribute. They can state that better resolution will help decide on the degree of disulphide bond participation is involved.

With the above now minor changes, I can support publication and trust the editor will follow up with the authors. My comments and concerns are very much in line with reviewer 2 so these changes need to be made.

Reviewer #2 (Remarks to the Author):

The manuscript by Zeng claimed to report real-time imaging of aggregation of single-stranded DNA triggered by disulfide bond chemistry. I appreciate the work that the authors did for the revision. In the revised manuscript, some earlier concerns of data analysis and over-interpretation of chemistry were addressed. I would recommend publishing after addressing the following minor revisions.

1. The electron dose $60 \text{ e}/(\text{Å}^2 \text{ s})$ is still quite high, although claimed to be low. It would be more convincing and also for readers to cross-compare the results with earlier work, especially those who also studied beam-sensitive materials, micelles, DNA, polymer, etc. Most of them used $1\text{-}10 \text{ e}/(\text{Å}^2 \text{ s})$, and with the use of radical scavengers which helps to retard beam-induced damage.
2. Authors might want to check with the plot for MSD. Fig S1, the plot indicates strong surface adsorption at the short time scale. The linear fits do not convey physical meaning here, if authors want to do linear fit, x-axis should be in log scale. Then the slope conveys the mechanism of diffusion.

Response to reviewers

Reviewer #1 (Remarks to the Author):

I appreciate the effort made to reply to my comments where my major concerns were related to the lack of adequate discussion of electron induced radical chemistry that would lead to multiple aggregation points not just at disulphide bonds. This point is covered. However, there is still a problem with the lack of appreciation of surface effects. In particular the statement "According to literature [25], the viscosity of liquid layer close to surface will increase 2 order of magnitude, thus caused much more slower diffusion dynamics than that in the bulk solution."

This statement is wrong. The cited paper is incorrect on this point or at least the authors should be more critical. How could the viscosity be that much higher? There is a slight propensity out to 3 waters in forming higher degree of H bonding of water at graphene surfaces...but not out many nm. Surface adhesion will give an effective increase in what would look like slow diffusive motion - even in considering the means square displacement as a function of time. There only have to be enough binding sites close enough to look diffusive in the spatial and temporal time resolution. This discussion must be tempered in which the role of the surface is given equal weight.

Response to reviewer: Thanks for the reviewer's effort on reviewing our manuscript. For the viscosity of liquid close to surface have been widely studied. These results support the claim of viscosity increase. We added more references about it here. For the viscosity of liquid layer near surface, there are papers reported that it will increase 2-6 order of magnitude, which caused by a layer of ordered liquid near the surface [1-4].

Andreas Verch et. al reported that [2], "The observed slow movement was not a result of hydrodynamic hindrance near a wall but instead explained by the presence of a layer of ordered liquid exhibiting a viscosity 5 orders of magnitude larger than a bulk liquid. The increased viscosity presumably led to a dramatic slowdown of the movement."

E R White et. al reported that [3], "Random displacements more than 3 orders of magnitude less than the expectation for free particles."

Svetlana Guriyanova et. al reported that [4], "The magnitudes of the effective viscosity measured in the experiment on gold surfaces are about 2 orders of magnitude smaller than those observed in pure water for native silicon oxide or for a surface terminated by hydroxyl groups (SiO₂-OH). The difference could be due to two reasons. First, and most important, is that the superficial viscosity depends strongly on the hydrophilicity. SiO₂-OH most probably is more hydrophilic than our gold electrodes. Second, in an AFM study, the maximum measured (or applicable) force is limited by the spring constant of the cantilever. Using a cantilever with a spring constant of 0.26 N/m, we probably could not drain the whole SVL from between tip and surface, so that the tip stops its approach before touching the gold surface."

[1] Haimei Zheng, Shelley A Claridge. Nanocrystal diffusion in a liquid thin film observed by in situ transmission electron microscopy. *Nano Lett.* 2009 Jun;9(6):2460-5.

[2] Andreas Verch, Marina Pfaff, Niels de Jonge. Exceptionally Slow Movement of Gold Nanoparticles at a Solid/Liquid Interface Investigated by Scanning Transmission Electron Microscopy. *Langmuir.* 2015 Jun 30;31(25):6956-64.

[3] E R White, Matthew Mecklenburg, Brian Shevitski, S B Singer, B C Regan. Charged nanoparticle dynamics in water induced by scanning transmission electron microscopy. *Langmuir.* 2012 Feb 28;28(8):3695-8.

[4] Svetlana Guriyanova, Victor G Mairanovsky, Elmar Bonaccorso. Supraviscosity and electroviscous effects at an electrode/aqueous electrolyte interface: an atomic force microscope study. J Colloid Interface Sci. 2011 Aug 15;360(2):800-4.

Final comment, the title needs to be changed as I still don't buy their argument on the disulphide bonds being the only source for aggregation.

Please change "Real-time Imaging of Disulfide Bond induced Aggregation of Single-stranded DNA" to "Real-time Imaging of Single-Strand DNA Aggregation".

Response to reviewer: Thanks reviewer for pointing it out. Combine reviewer's suggestion, and consider that the single stranded DNA that we're using is sulfhydryl modified, we revised the title as follows: Real-time Imaging of Sulfhydryl Single-stranded DNA Aggregation.

The authors should state throughout the manuscript that the observations are consistent/correlated with disulphide bond cleavage and aggregation but there are also other possible radiolysis processes that likely also contribute. They can state that better resolution will help decide on the degree of disulphide bond participation is involved.

Response to reviewer: Thanks reviewer for pointing it out. We have added these statements in the line 116-120, as followed: However, better resolution of LP-TEM will help deciding the degree of disulphide bond participation.

With the above now minor changes, I can support publication and trust the editor will follow up with the authors. My comments and concerns are very much in line with reviewer 2 so these changes need to be made.

Reviewer #2 (Remarks to the Author):

The manuscript by Zeng claimed to report real-time imaging of aggregation of single-stranded DNA triggered by disulfide bond chemistry. I appreciate the work that the authors did for the revision. In the revised manuscript, some earlier concerns of data analysis and over-interpretation of chemistry were addressed. I would recommend publishing after addressing the following minor revisions.

1. The electron dose $60 \text{ e}/(\text{Å}^2 \text{ s})$ is still quite high, although claimed to be low. It would be more convincing and also for readers to cross-compare the results with earlier work, especially those who also studied beam-sensitive materials, micelles, DNA, polymer, etc. Most of them used $1-10 \text{ e}/(\text{Å}^2 \text{ s})$, and with the use of radical scavengers which helps to retard beam-induced damage.

Response to reviewer: Thanks reviewer for pointing it out. The electron beam dose rate distribution reported in previous articles ranges from 0.21 to $100 \text{ e}/(\text{Å}^2 \cdot \text{s})$ [1-4]. Here we use $60 \text{ e}/(\text{Å}^2 \cdot \text{s})$ as an intermediate dose. And we adopt PBS system as solvent for DNA, which has high buffering capacity. The H^+ and OH^- produced by electron beam irradiation will be neutralized and consumed, thus reducing radiation damage. We have added this discussion in to main text in line 78-81.

Chen et al. also reported [1] prolonged stability of the nanoconjugate structure composing of dsDNA linkers throughout the observation under an estimated dose rate of 60–100 $e^-/(\text{\AA}^2\cdot\text{s})$, which is more than the critical dose rate previously reported for biological molecules.

Xiao Xie et al. reported that [2] they use Liquid-Cell Scanning Transmission Electron Microscopy to characterize DNA-Directed Gold Nanoparticle Assemblies with dose rate from 0.21 to 3.75 $e^-/(\text{\AA}^2\cdot\text{s})$.

Chen et al. reported that [3] they Visualize Multimerization and Self-Assembly of DNA Functionalized Gold Nanoparticles Using In-Liquid Transmission with dose rate from 1.1 to 1.4 $e^-/(\text{\AA}^2\cdot\text{s})$.

Huan Wang reported that [4] they use liquid-cell electron microscopy captured the intermediate states of DNA self-assembly with dose rate from 2 to 10 $e^-/(\text{\AA}^2\cdot\text{s})$.

[1] Qian Chen et al. [1]. Nano Lett. 2013 Sep 11;13(9):4556-61.

[2] Sercan Keskin, Stephanie Besztejan et al. Visualization of multimerization and self-assembly of DNA-functionalized gold nanoparticles using in-liquid transmission electron microscopy. J Phys Chem Lett. 2015 Nov 19;6(22):4487-92.

[3] Xiao Xie, Jing Tu, et al. Nanoscale Observation of Conformational Transformation of DNA Polymerase via In-Situ Liquid-Cell Transmission Electron Microscopy. J Biomed Nanotechnol. 2019 May 1;15(5):1106-1111.

[4] Huan Wang, Bo Li, et al. Intermediate states of molecular self-assembly from liquid-cell electron microscopy. Proc Natl Acad Sci U S A. 2020 Jan 21;117(3):1283-1292.

2. Authors might want to check with the plot for MSD. Fig S1, the plot indicates strong surface adsorption at the short time scale. The linear fits do not convey physical meaning here, if authors want to do linear fit, x-axis should be in log scale. Then the slope conveys the mechanism of diffusion.

Response to reviewer: Thanks reviewer for pointing it out. In the old version of the manuscript, we did not consider drift in the MSD analysis. In the new version, we corrected the drift of the images and re-counted the MSD of individual particles instead of the average of several particles with extended the statistical time showed in Supplementary Figure S1, we found that they were in random Brownian motion. In reference to the reviewer's comments, we deleted the linear fits.